# A Greek Translation of the Brunel Mood Scale: Initial Validation among Exercise Participants and Inactive Adults

**DOI:** 10.3390/sports11120234

**Published:** 2023-11-27

**Authors:** Symeon P. Vlachopoulos, Andrew M. Lane, Peter C. Terry

**Affiliations:** 1Laboratory of Social Research on Physical Activity, Department of Physical Education and Sport Science at Serres, Aristotle University of Thessaloniki, 62122 Serres, Greece; vlachop@phed-sr.auth.gr; 2School of Psychology, Psychological Research Centre, University of Wolverhampton, Walsall WS1 3BD, UK; 3School of Psychology & Wellbeing, Centre for Health Research, University of Southern Queensland, Toowoomba, QLD 4350, Australia; peter.terry@unisq.edu.au

**Keywords:** Greece, mood, measurement, exercise, physical activity, validity

## Abstract

The aim of the present study was to provide initial validity evidence of a Greek translation of the 24-item Brunel Mood Scale, referred to as the BRUMS-Greek, a measure of anger, confusion, depression, fatigue, tension, and vigour. Data were collected from 1417 Greek adult exercise participants and 369 physically inactive adults, totaling 1786 adults (male = 578, female = 1208) aged 18–64 years (*M* = 34.73 ± 11.81 years). Given the large univariate and multivariate non-normality, a confirmatory factor analyses treating responses as ordered categorical variables was conducted which supported the hypothesised six-correlated factor measurement model. The internal consistency reliability of the BRUMS-Greek subscales was supported via Cronbach alpha coefficients. The construct validity of the scales was supported (a) via correlations in the hypothesised direction with trait positive and negative affect, (b) with more positive and less negative moods reported immediately after participation in a single exercise class compared to pre-exercise mood, and (c) with exercise participants reporting more positive and less negative mood states compared to physically inactive adults. Women reported higher tension and lower vigour scores than men. Tension scores were higher and confusion scores lower among younger participants (≤35 years) than older participants (≥36 years). Participants with obesity reported higher negative mood scores than those who were underweight or normal weight. In sum, the BRUMS-Greek demonstrated acceptable psychometric characteristics, and is proposed to be a suitable measure for use with exercise participants, physically inactive adults, and other Greek populations to explore research questions related to mood.

## 1. Introduction

Investigation of the role of mood in sports performance [1,2] and exercise behaviour [3,4,5] has been longstanding in the sphere of sport and exercise psychology [6,7]. In the present investigation, mood is defined as “a set of feelings, ephemeral in nature, varying in intensity and duration, and usually involving more than one emotion” (p. 17, [8]). Moods are viewed as having a valence dimension varying from positive (e.g., happy) to negative (e.g., depressed) and an arousal dimension varying from activation (e.g., alert) to deactivation (e.g., tired) [9]. Regarding the distinction between moods and emotions, moods are seen as more diffuse, of lesser intensity and longer duration, and not related to a specific cause [10,11]. Persistent and/or extreme negative moods may mirror increased risk of mental health disorders [12]. The measure of preference for a large proportion of the research conducted on mood in sport and exercise over several decades has been the Profile of Mood States [13,14] and its derivatives.

The 24-item Brunel Mood Scale (BRUMS) was developed as a shortened version of the POMS designed to be suitable for mood assessment in adolescents as well as adults [15,16]. The BRUMS has a shorter completion time (2–3 min) compared to the 65-item POMS (7–10 min) and is validated for both adolescents and adults, including athletes. The BRUMS contains six subscales representing the mood dimensions of anger, confusion, depression, fatigue, tension, and vigour. It should be noted that the depression score is an indicator of depressed mood rather than clinical depression [16]. The POMS was originally developed for use with psychiatric outpatients and has been criticised for its negative orientation and for providing a limited assessment of mood rather than a comprehensive measure of the mood construct [17]. Nevertheless, the POMS has shown utility in screening for mental health issues [6] and predictive effectiveness in competitive domains including sports, where negative moods can hinder performance [1,2]. The BRUMS has often been used to predict sports performance, and also to assess athletes’ mood responses to various situational stressors, such as training load, underperformance, injury [18], poor sleep quality [19], rapid weight loss [20,21], and restricted food and drink intake during Ramadan [22].

Lane and Terry [8] introduced a conceptual model hypothesising the interactive effects of different mood dimensions on performance, focusing particularly on how depressed mood interacts with anger and tension. It has been demonstrated that in the absence of symptoms of depressed mood, anger has been positively associated with good performance [23,24] while under evident symptoms of depressed mood, anger has been linked to poor performance [1]. Negative mood has also been found to have positive effects on some aspects of performance, such as increasing cognitive and emotional creativity by enhancing self-focused attention [25]. In the exercise domain, the BRUMS has been used to assess mood responses to various modes of exercise [26] and to monitor mood responses to music [27]. Outside of the sport and exercise domains, the BRUMS has been used, for example, to screen for post-traumatic stress disorder among military personnel [28], monitor psychological well-being among cardiac rehabilitation patients [29], assess adolescents with elevated suicide risk [30], evaluate the effects of active video games among children [31], test the effects of social dance sessions among people with Parkinson’s disease [32], gauge mental health risk among higher degree by research students [33], and quantify mood changes associated with the COVID-19 pandemic [34,35,36].

Mood profiling, a process in which scores on a mood scale are plotted against normative scores to create a graphical profile, has been used extensively to identify common patterns of mood states using either the POMS [1,2,6] or the BRUMS [37]. Several distinct patterns of mood have been detected using the BRUMS [37]. For instance, the iceberg profile, which is defined by a high vigour score (greater than the mean standardised score of 50) combined with low scores for tension, depression, anger, fatigue, and confusion (lower than the mean standardised score of 50) is typically linked to positive mental health and high athletic performance, as initially emerged using the POMS [1,2,6,38]. The inverse iceberg profile, characterised by a below-average vigour score combined with above-average tension, depression, anger, fatigue, and confusion scores, generally associates with reduced performance and increased risk of pathogenesis, as found using the POMS [39] or a standardized 40-item mood scale [40]. The inverse Everest profile, the most negative profile, characterised by a low vigour score, high scores for tension and fatigue, and very high scores for depression, anger, and confusion as assessed by the BRUMS, indicates an elevated risk of clinical conditions such as post-traumatic stress disorder [28]. The shark fin profile, reflecting below-average tension, depression, anger, vigour, and confusion in combination with very high fatigue scores assessed by the BRUMS, may signal an increased risk of athletic injury [41]. The surface profile consists of average scores on all mood dimensions and represents an average mood [37], and the submerged profile has below-average scores on all mood dimensions, with both profiles assessed by the BRUMS [37]. The submerged profile was reported among support staff of the Irish Olympic team involved in long-haul transmeridian travel for the 2020 Olympic Games in Japan [42], which was interpreted as sub-optimal for their role, although a submerged profile may prove beneficial in sports such as pistol shooting, where good performance depends on remaining calm and unemotional [43].

The BRUMS has previously been validated in at least 15 languages other than English [44,45,46,47,48,49,50,51,52,53,54,55,56,57,58] and in a Singaporean context [59]. To date, there has been no translation of the scale into the Greek language, despite it being spoken by over 13 million people worldwide (https://en.wikipedia.org/wiki/Greek_language, 21 November 2023). Cross-cultural comparison is a valuable tool for researchers to test the external validity and generalisability of their theories, measures, and models [60,61]. Accordingly, Duda and Allison [62] expressed a plea to researchers in sport and exercise psychology to incorporate the variables of culture and ethnicity into their research agendas. Therefore, translating and validating instruments to a non-English language may contribute to testing the globality and generalisability of theory outside of English language boundaries.

The aim of the present study was to examine the validity aspects of a Greek translation of the BRUMS, referred to as the BRUMS-Greek, to facilitate research involving the mood construct among Greek-speaking populations. We hypothesised that the 24-item translated-into-Greek BRUMS scores would support a six-correlated factor measurement model in line with previous translated versions of the scale [44,46,51,52,53,54], and that the BRUMS-Greek factors would display adequate internal consistency reliability [63]. We also tested the concurrent validity of the BRUMS scores using the constructs of trait positive affect and negative affect, hypothesising that vigour would positively correlate with the trait positive affect and negatively with the trait negative affect. Inversely, positive correlations were expected between the BRUMS negative mood subscales and the trait negative affect, and negative correlations with the trait positive affect.

To assess the known-groups validity [64,65], we analysed between-group differences on several variables previously found to be associated with mood differences. For sex differences, we hypothesised that males would report more positive and less negative moods than females. Cañadas et al. [57] and Han et al. [59] found that men reported more positive and less negative moods compared to women, and women typically report feeling more depressed, sad, anxious, nervous, fatigued, and lacking interest or energy, than men [66]. In terms of age group comparisons, it was hypothesised that older participants would report higher vigour scores and lower scores on unpleasant mood states compared to younger adults. Older individuals have been found to report lower confusion, fatigue, and tension compared to younger persons [59] with older athletes also reporting lower confusion, depression, fatigue, and tension compared to younger athletes [52]. Regarding the body mass index (BMI) categories, it was hypothesised that a greater BMI would correspond to more negative moods. Obesity has been theorised to be comorbid with mood disorders [67] and has also been linked to symptoms of depression [68].

We assessed mood changes from pre- to post-exercise, investigating the main effects for changes in mood over time, and interaction effects for age, sex, and BMI. Given the positive effects of physical exercise on moods in non-clinical populations [69,70], it was hypothesised that BRUMS responses in a sample of Greek-speaking adult exercise participants would be more positive and less negative immediately after participation in a single, group-based, aerobic indoor exercise class compared to their pre-exercise mood, and that exercise participants would report more positive and less negative mood states compared to physically inactive adults. For the interaction effects, our approach was more exploratory. If women, younger participants, and individuals with obesity reported a more negative mood before exercise, perhaps they would report greater mood enhancements than male, older, and normal-weight participants post exercise, given that Lane and Lovejoy [71] have shown exercise to be particularly effective at enhancing mood among participants who reported a negative mood at baseline. Testing the interactions between exercise and sex, exercise and age, and exercise and BMI not only adds useful information about the utility of the BRUMS-Greek, but also extends knowledge about the value of exercise as a method to enhance mood. Therefore, the main objectives of the study were to provide initial validity evidence for the BRUMS-Greek, to examine links between mood scores and the constructs of trait positive and negative affect, and to advance the use of the BRUMS instrument in relation to exercise behavior.

## 2. Materials and Methods

### 2.1. Participants

A heterogeneous nonprobability sample of 1786 Greek adults was studied, comprising 1417 exercise participants (79.3%) and 369 physically inactive adults (20.7%). Demographic information in terms of sex, age, height, weight, body mass index, and exercise participation is provided in Table 1. Of the 1417 exercise participants, 785 reported participating in group-based indoor exercise programs (55.4%) while 632 were involved in individual athletic activities (44.6%). Individuals were defined as exercise participants if they attended an exercise program in a fitness center. Individuals were defined as inactive if they did not exercise at all for a period of at least 12 consecutive months.

### 2.2. Measurement of Mood

Mood was assessed using a Greek translation of the Brunel Mood Scale (BRUMS), a 24-item instrument developed to measure mood among adolescents and adults [15,16]. The BRUMS was adapted from the Profile of Mood States [13,14] and includes six subscales of four items each, measuring tension (nervous, anxious, worry, panicky), depression (unhappy, miserable, depressed, downhearted), anger (annoyed, bitter, angry, bad tempered), vigour (energetic, active, lively, alert), fatigue (exhausted, tired, worn out, sleepy) and confusion (mixed up, muddled, uncertain, confused). Responses are provided on a 5-point Likert-type scale (0 = not at all, 1 = a little, 2 = moderately, 3 = quite a bit, 4 = extremely) with subscale scores, ranging from 0 to 16, created by summing the four items. Responses are provided to the question “How do you feel right now?” Cronbach alpha coefficients for all subscales have generally ranged within acceptable levels between 0.74 and 0.90 [15,16]. In the original validation studies, the BRUMS demonstrated adequate psychometric properties using multi-group confirmatory factor analysis supporting configural, metric, scalar, and residual invariance across adult students, adult athletes, young athletes, and school children [15,16]. In addition to producing six mood subscale scores, an overall Total Mood Disturbance (TMD) score may be calculated by summing the scores for tension, depression, anger, fatigue, and confusion and then subtracting the vigour score. However, calculation of the TMD score is not generally recommended [15,16] because (a) combining six scores into one results in an unnecessary loss of information and (b) in the context of the TMD, anger and tension are inherently treated as negative mood states, whereas research has shown them to be facilitative, depending on their interaction with depression [8,24] and in combat sports [23].

### 2.3. Measurement of Affect

Trait affect was assessed as a concurrent measure using the 10-item International Positive and Negative Affect Schedule—Short Form (I–PANAS-SF) [72], a derivative of the original PANAS [73]. For negative affect, descriptors were upset, hostile, ashamed, nervous, and afraid. For positive affect, descriptors were alert, inspired, determined, attentive, and active [72]. Responses were provided to the question “Thinking about yourself and how you normally feel, to what extent do you generally feel” on a 1–5 scale anchored by never (1) and always (5). Favorable psychometric support has been obtained regarding cross-sample stability, internal reliability, temporal stability, cross-cultural factorial invariance, convergent and criterion-related validity [72].

### 2.4. Procedure

The translation-back translation method [74] was used to translate the BRUMS and the I–PANAS–SF into Greek. The instruments were initially translated into Greek and then back into English by two bilingual (Greek and English) researchers holding a PhD in sport and exercise psychology and sports science, respectively. When congruence was secured in item meaning between the original and back-translated English versions, the Greek translation was retained. The Greek translation of the Brunel Mood Scale is referred to as the BRUMS-Greek and the Greek translation of the International Positive and Negative Affect Schedule—Short Form is referred to as the I–PANAS–SF–Greek.

Regarding data collection, exercise participants were approached in fitness centers in northern Greece. Initially, verbal permission was granted by the directors of the fitness centers to contact the participants at the reception area. Participants were informed about the nature and objectives of the study. The purpose of the study was explained to them and they were informed that their participation in the study was optional and that they could discontinue participation at any time. Participation in the study was voluntary without any incentives provided. The questionnaires to be completed were distributed before participation in an exercise class. Those who agreed to participate provided their written informed consent and completed the questionnaire in a quiet area close to the reception. Questionnaire completion took approximately 10 min and was supervised by a research assistant.

Participants of the physically inactive subsample were recruited using snowball sampling [75]. Both samples were non-probability samples. Participants were treated in accordance with the American Psychological Association (APA) ethical guidelines and were reassured that their responses would be kept confidential and anonymous. The study complied with the Code of Ethics of the World Medical Association Declaration of Helsinki. The protocol received approval from the research ethics committee of the Department of Physical Education and Sport Science at Serres, Aristotle University of Thessaloniki in Greece (Approval #ERC-018/2020).

A subsample of 369 physically inactive adults also completed the I-PANAS-SF-Greek to examine the concurrent validity of the BRUMS-Greek scores. Another subsample of 398 exercise participants was measured before and immediately after participation in a single group-based indoor exercise class to compare BRUMS-Greek scores pre- and post-exercise. Demographic data (sex, age, height, weight, participation/non-participation in exercise) were also collected.

### 2.5. Data Analysis

Initially, descriptive statistics were calculated for individual items and subscales of the BRUMS. The measurement model of the BRUMS was examined via Confirmatory Factor Analysis (CFA) using the Maximum Likelihood Method of estimation in EQSWIN 6.1 [76]. CFA is a theory-driven method in the context of which several estimation techniques and goodness-of-fit indices are used to examine how well a hypothesised model fits the sample covariance matrix. The rule of 10 participants per parameter to be estimated was followed [77]. Such a rule corresponds to a sample size of at least 630 participants. We specified that items were related only to their hypothesised factor. The mood factor variances were fixed to 1.0 and item error covariances were fixed to a value of zero. According to Hu and Bentler [78], ML is less sensitive to distribution misspecification and performs well over other normal theory-based methods for large samples [79]. The latent factors of anger, confusion, depression, fatigue, tension, and vigour were allowed to intercorrelate [47,49,51].

The goodness of fit index initially considered was the *χ2/df* ratio where a value < 3 indicates an acceptable model fit [80]. However, because the χ^2^ test is difficult to satisfy with sample sizes larger than 200 [80], this index was not given priority for the present sample of 1786. Instead, the Comparative Fit Index (CFI) [81] was first considered. CFI values close to 0.95 indicate an excellent fit to the data [82] whereas values of 0.90 or greater denote an acceptable fit. Additionally, the Root Mean Square Error of Approximation (RMSEA) [83] along with its 90% confidence interval was used. The RMSEA reflects the mean discrepancy between the observed covariances and model-implied covariances per degree of freedom with a value lower than 0.05 reflecting a good model fit [82], and values up to 0.08 representing an adequate fit [84,85]. The models tested were the original six-correlated factor model [16] and a single-factor model (where all BRUMS items were specified to load onto a single factor) for comparison purposes. Akaike’s information criterion (AIC) was used to compare the competing models because it penalises for model complexity [85]. Smaller AIC values denote a better model fit. Internal consistency reliability of the BRUMS subscales was evaluated using Cronbach α [63].

Pearson’s correlations were computed on the subsample of 369 physically inactive individuals to examine concurrent validity associations of I-PANAS-SF trait positive and trait negative affect with BRUMS-Greek subscale scores. A repeated measures MANOVA was computed to examine mood differences for a subsample of 398 exercise participants before and immediately after participation in a single exercise class. One-way MANOVAs were computed to compare exercise participants (*n* = 1417) with physically inactive participants (*n* = 369) and examine potential differences on BRUMS-Greek scores across participant sex, age group, and BMI category. In instances where multivariate effects were shown to be significant, univariate F-tests were used to identify the source of differences among the subscales. A Bonferroni adjustment was applied to the alpha level (0.05/6 = 0.008) to account for our six dependent variables (i.e., anger, confusion, depression, fatigue, tension, and vigour) [79]. Additionally, to establish a preliminary table of normative data for use with Greek-speaking healthy adults, raw scores on each BRUMS-Greek subscale were converted to T-scores using the formula: T = 50 + 10z [79].

## 3. Results

Data were screened to examine compliance with univariate and multivariate normality. Significant univariate non-normality was found for the subscale scores (especially for anger, confusion, and depression), and specific items of all negative moods (anger, confusion, depression, tension, and fatigue) while univariate normality was found for the vigour subscale and items scores. Such non-normality is consistent with typical mood subscale distributions [15,16]. Excessive multivariate non-normality was also evident for the set of BRUMS-Greek items (normalised estimate of Mardia’s coefficient of multivariate kurtosis = 335.05, that is >5.00) [85]. However, in previous validation studies of the BRUMS [37,49,54,59], non-normality was also observed and a good model fit was achieved without data transformations. Also, Nevill and Lane [86] proposed that self-report measures should not be transformed because measurement scales operate at an interval rather than a ratio level. For this reason, no data transformations occurred prior to analysis. Thirteen multivariate outliers were identified using a Mahalanobis distance test (*p* < 0.001). A case-by-case inspection showed no evidence of response bias in the form of extreme, acquiescent, or straight-line responding [87,88]. Therefore, all outliers were retained and a sample of 1786 cases was analysed.

### 3.1. Confirmatory Factor Analysis of the BRUMS-Greek

Due to the multivariate non-normality of the data, the ML robust method of estimation using EQSWIN 6.1 [76] was used. This method provides the non-normality corrected Satorra–Bentler Scaled χ^2^ (S-B χ^2^), CFI, RMSEA, and its 90% CI (called robust estimates). Additionally, high values of individual item skewness and kurtosis in several BRUMS items led to treating the variables as ordered categorical rather than continuous. Therefore, a polychoric correlation matrix was analysed. The CFI has been found to be least affected by non-normality when treating data as ordered categorical [89]. The CFA results supported a good fit for the six-correlated factor model of the BRUMS-Greek scores: S-B scaled *χ2* = 1281.39, *df* = 237, *p* < 0.001, robust CFI = 0.961, robust RMSEA = 0.050 (90% CI = 0.047–0.052), and robust AIC = 807.39, whereas the single-factor model did not fit the data well [S-B scaled *χ2* = 3077.74, *df* = 252, *p* < 0.001, Robust CFI = 0.895, Robust RMSEA = 0.079 (90% CI = 0.077–0.082), and robust AIC = 2573.74]. The fully standardised item loadings ranged from 0.663–0.919. The mean scores, standard deviations, skewness, and kurtosis values of the BRUMS-Greek items are presented in Table 2. Subscale descriptive statistics and Cronbach alpha values are presented in Table 3. All alpha values were greater than 0.70. Subscale inter-correlations between the negative mood subscales were positive, and the negative mood subscales were inversely correlated with vigour (Table 3).

### 3.2. Confirmatory Factor Analysis of the I-PANAS-SF-Greek

As we used a Greek version of the I–PANAS–SF to examine the concurrent validity of the BRUMS-Greek subscales, we first computed a CFA to examine the factor structure of the I-PANAS-SF-Greek. Variables were treated as ordered categorical. A two-correlated factor model was specified and tested. Noting the multivariate non-normality of the data, the ML robust method of estimation [76] was used. The fit indexes supported an adequate fit of the two-correlated factor measurement model of the I-PANAS-SF-Greek: S-B scaled *χ^2^* (*n* = 369) = 94.49, *df* = 34, *p* < 0.001, Robust CFI = 0.915, Robust RMSEA = 0.070, and (RMSEA 90% CI = 0.053, 0.086). Item loadings ranged from 0.533 to 0.873. The latent factor correlation was −0.28 (*p* < 0.05) while the Pearson correlation between subscale scores was −0.18 (*p* < 0.001).

### 3.3. Concurrent Validity

Correlations between the BRUMS-Greek subscale scores and the trait positive affect and negative affect scores of the I-PANAS-SF-Greek aligned with theoretical expectations. Trait negative affect was positively correlated with the negative BRUMS-Greek subscales scores and negatively correlated with the vigour scores. Conversely, trait positive affect was negatively correlated with the negative BRUMS-Greek subscale scores and positively correlated with the vigour scores, thereby supporting the concurrent validity of the BRUMS-Greek scale (Table 4).

### 3.4. Group Differences in Mood

A one-way repeated measures MANOVA was computed to compare mood scores before and immediately after participation in a single group-based exercise class among a subsample of 398 exercise participants. A significant multivariate effect was found [Hotelling’s *T* = 0.330, *F* = 21.47 (6, 390), *p* < 0.001, and partial eta squared = 0.248] that explained 24.8% of the variance. Univariate *F*-tests indicated significant pre- to post-exercise differences for all moods except fatigue (Table 5). Lowered negative moods were reported after exercise compared to before exercise, whereas vigour scores increased after exercise. The strongest effect was found for tension scores, which showed a medium effect size. Other effect sizes were small, apart from fatigue, which showed a very small effect (Table 5).

A one-way MANOVA comparing exercise participants (*n* = 1417) with physically inactive adults (*n* = 369) by mood scores similarly showed a significant multivariate effect [Hotelling’s *T* = 0.077, *F* = 22.95 (6, 1779), *p* < 0.001, and partial eta squared = 0.072], explaining 7.2% of variance. Univariate *F*-tests revealed significant differences for all mood subscale scores except vigour with exercise participants reporting lower scores for negative moods (Table 5). Hedge’s *g* effect sizes [79] identified medium-sized differences for depression, fatigue, and tension scores, small differences for anger and confusion, and a very small effect for vigour (Table 5).

Male and female participants were also compared using a one-way MANOVA, which identified a significant multivariate effect [Hotelling’s *T* = 0.022, *F* = 6.45 (6, 1779), *p* < 0.001, and partial eta squared = 0.021] explaining 2.1% of variance. Univariate *F*-tests showed significant effects for tension and vigour, with women reporting higher tension scores, and lower scores for vigour. All of the Hedge’s *g* effect sizes were below 0.20 indicating very small effects (Table 5).

Mood scores of participants younger and older than 35 years were compared using a one-way MANOVA. This age cut-off was used to create similar-sized groups. Results showed a significant multivariate effect [Hotelling’s *T* = 0.019, *F* = 5.74 (6, 1779), *p* < 0.001, and partial eta squared = 0.019] that explained 1.9% of the variance. Univariate *F*-tests indicated significant differences for confusion and tension, with older participants reporting lower tension scores but higher confusion scores. Hedge’s *g* indicated a small effect for confusion and very small effects for the remaining mood subscales (Table 5).

In consideration of the literature linking obesity (rather than overweight) to mood disorders [67] and symptoms of depression [68], a one-way MANOVA was computed to compare mood scores of underweight and normal-weight participants with a second group comprising only those participants with obesity and excluding the overweight participant scores. Results showed a significant multivariate effect [Hotelling’s *T* = 0.014, *F* = 3.05 (6, 1280), *p* < 0.01, and partial eta squared = 0.014], explaining 1.4% of variance. Univariate *F*-tests revealed a significant, but small difference for fatigue scores. For all mood subscales (significant and nonsignificant effects), those with obesity reported higher negative mood scores and lower vigour scores (Table 5). Two-way repeated measures MANOVAs were conducted to explore the interactions effects between the pre- and post-exercise mood changes by sex, age, and BMI category, but none of the interactions were significant.

To assist researchers and practitioners wishing to use the BRUMS-Greek, we constructed a preliminary table of normative data showing raw scores and the equivalent T-score (Table 6) based on data from the overall sample. We also produced a profile sheet to allow raw scores to be plotted against the norms to produce individual or group mood profiles (Figure 1). The BRUMS-Greek questionnaire and scoring instructions are shown in Figure 2.

## 4. Discussion

The aim of the present study was to extend validity evidence of the Brunel Mood Scale [15,16] outside the English language boundaries using a Greek translation of the scale, referred to as the BRUMS-Greek. We evaluated the factorial validity, internal consistency reliability, concurrent validity, and construct validity of the BRUMS-Greek using a heterogeneous sample of 1786 Greek adults including exercise participants and physically inactive adults.

### 4.1. Factorial Validity and Internal Consistency Reliability

Using the rule of 10 participants per parameter to be estimated [77], and treating variables as ordered categorical owing to the non-normality of several items, mainly across the negative mood factors, confirmatory factor analysis supported a good model fit for the hypothesised six-correlated factor measurement model, whereas an alternative single-factor measurement model was not supported. All item loadings were strong (>60) and statistically significant, meaning that the translated items functioned as efficient indicators of their intended mood factor. The BRUMS-Greek subscale inter-correlations supported theoretical predictions with negative moods being significantly and positively intercorrelated, and negatively correlated with vigour (Table 3). The substantially lower magnitude of correlations of vigour with the negative subscales, compared to the strong inter-correlations among the negative subscales, supports the proposed conceptual independence of the vigour construct [15,90]. Cronbach alpha coefficients supported the internal consistency reliability for all of the BRUMS-Greek subscales, with alpha values greater than 0.70. The present findings are consistent with previous BRUMS validation studies, including among Italian sports participants [52], Bangladeshi participants [45], Singaporean athletes and non-athletes [59], Malaysian athletes [55,91], and the original scale development work of the BRUMS [15,16].

### 4.2. Concurrent Validity

Pearson correlations of the BRUMS-Greek subscale scores with trait positive affect and negative affect scores supported concurrent validity. Negative BRUMS-Greek subscale scores were significantly and positively correlated with trait negative affect and inversely correlated with trait positive affect. The reverse was true for vigour scores. These results are consistent with previous findings that similarly identified significant relationships in the hypothesised direction between BRUMS subscale scores and measures of positive and negative affect, depression, stress, and anxiety [52,59], mental health [45], and perceived stress and neuroticism [54]. Our results also demonstrated the factorial and concurrent validity of a Greek language version of the I-PANAS-SF [72] and therefore support the validity of both the BRUMS-Greek and I-PANAS-SF-Greek scales.

### 4.3. Between-Group Differences

Comparison of the BRUMS-Greek scores before and immediately after participation in a single group-based exercise class identified significantly improved mood after the exercise session on all subscales except fatigue. Previous research has also demonstrated the positive effects of exercise participation on moods in non-clinical populations [69,70]. Herring and O’Connor [69] found that acute moderate-to-high intensity lower-body resistance exercise increased feelings of energy during and after exercise among young sedentary women reporting below-average feelings of energy immediately prior to the resistance exercise bout. However, as in our study, no significant change in fatigue was reported post-exercise. The exercise participants of this subsample attended indoor group-based exercise programs combining aerobic and resistance exercises, mainly of a moderate-to-high intensity, and therefore it is reasonable to expect high fatigue scores post-exercise [26]. It appears plausible that the physical exertion involved in moderate-to-high intensity exercise, which would logically generate increased feelings of fatigue, was offset by the mood-enhancing benefits inherent in exercise. Various forms of aerobic exercise have been shown to enhance mood [26] and persistent enhancement of mood is promoted by participation in long-term exercise [92]. Moreover, there is compelling evidence of the benefits of physical exercise for the treatment and prevention of mental health issues, particularly depression and anxiety [70,93,94]. Hence, the present findings that exercise participants reported significantly lower scores on all negative subscales than physically inactive participants supported the known-group validity [65], and therefore the construct validity, of the BRUMS-Greek measure. The exact mechanisms by which the beneficial effects of exercise on mood occur are not well understood, although an increase in blood circulation to the brain that influences the hypothalamic-pituitary-adrenal (HPA) axis and reduces reactivity to stress, plus the social interaction, self-efficacy, and distraction effects of exercise have been offered as potential explanations [95].

Regarding differences in participant sex, women reported significantly higher tension and lower vigour than men, albeit with very small effect sizes. In previous BRUMS studies, men have similarly been shown to report more positive and less negative moods compared to women [57,59,96] while women have generally been found to report feeling more depressed, sad, anxious, nervous, fatigued, and lacking energy more than men [66]. Explanations for mood differences between men and women have often been explained by biochemical [97] and neurological factors [98], although there are clear sociological differences between the sexes that would account for mood differences, perhaps the most obvious being the enduring disadvantage experienced by women in workplace, domestic, and educational settings [99,100,101]. The extent of mood differences between men and women in the present study is very modest compared to those reported in larger-scale studies using the BRUMS [96], suggesting that the BRUMS-Greek measure could be applied in future studies to further explore mood differences among males and females in a Greek-language context among various sporting groups and the general Greek population.

Regarding age group differences, individuals up to 35 years reported lower confusion and higher tension scores than those 36 years and above, albeit with small or very small effect sizes. Previous research using the BRUMS has tended to show significantly lower negative mood scores among older age groups [52,59,96]. For example, Han et al. [59] found higher confusion, fatigue, and tension scores reported by younger participants, while younger athletes reported greater confusion, depression, fatigue, and tension than their older counterparts [52]. The higher levels of tension reported by the younger age group are consistent with previous research, whereas their lower confusion is not [96]. The higher confusion scores reported by the older age group and the absence of age-related differences in depression and fatigue found in previous studies may be anomalous or may reflect some characteristic of Greek participants that is not shared by participants in other cultural contexts [52,59,96]. Future research conducted using the BRUMS-Greek should explore age-related differences in moods further, noting that mood disorders typically emerge before the age of 30 [102].

A comparison of the mood scores of participants who were categorised by their BMI as being underweight/normal weight with those participants categorised as obese showed that the latter reported more negative moods generally with significantly higher scores for fatigue, although effects were small or very small in magnitude. Previous evidence has indicated that obesity and mood disorders share a series of clinical, neurobiological, genetic, and environmental factors [67]. Further, Frank et al. [68] in a multi-cohort study, found that obesity was robustly associated with physical (could not get going/lack of energy), cognitive (reduced interest in doing things), emotional (feeling depressed), and self-perception (feeling bad about oneself) symptoms of depression. Although the present results are consistent with previous findings related to BMI categorisations, there is scope for future research using the BRUMS-Greek in this area, such as assessing if mood enhancements occur when moving from the obese category into a healthier weight category and conversely whether mood decrements are apparent among those moving from a normal weight to an overweight category. In a sporting context, particularly in weight-restricted events such as boxing, judo, and rowing, failure to make weight would be hypothesised to have an immediate and significant detrimental impact upon mood.

### 4.4. Implications of the Findings

The POMS, and by implication the BRUMS, have been criticised for being atheoretical [103] and for assessing primarily negative mood states [7]. Also, Ekkekakis [17] suggested other limitations, including the proposed conceptual advantages of using bipolar rather than unipolar scales in measuring mood, and the need to avoid generalisations based on the six distinct mood states to the global mood domain. Therefore, it should be noted that the BRUMS, in English or in translation, is a measure of six distinct mood dimensions rather than a comprehensive measure of the mood construct. However, unipolar measures of mood such as the POMS and the BRUMS have been found to be more sensitive indicators of well-being compared to objective physiological measures such as salivary biomarkers and blood [104]. Moreover, mood profiles, especially the inverse iceberg and inverse Everest profiles, have been used in the sports science and sports medicine areas to provide indicators of increased risk of injury [41], overtraining syndrome [39,46,105], and poor adaptation to training load [106,107].

Research on mood in the Greek language has to date been somewhat limited, although investigations have been conducted into, for example, the impact of exercise on mood among yoga participants [108], cancer patients [109], and prison inmates [110]. Mood-related research in the Greek language has been restricted by the lack of relevant validated measures. A version of the POMS in Greek is mentioned in a paper by Roussi and Vassilaki [111] but few details of the scale are provided, and an unpublished 30-item Greek version of the POMS was produced by Zervas and colleagues in 1993 [112]. Development of the BRUMS-Greek facilitates many applied and research opportunities among Greek-speaking populations. In particular, the BRUMS-Greek could be used to investigate whether the six distinct mood profile clusters (i.e., iceberg, inverse Everest, inverse iceberg, shark fin, submerged, and surface profiles) that have been identified in English-speaking populations [18,37] and in other languages [113,114] and cultural contexts [115] are also evident among Greek-speaking populations, and their relative prevalence.

From an applied perspective, the measure has many uses in the sports domain. Given the applications previously reported in the literature [46,105,116], the BRUMS-Greek may have utility (a) as a self-monitoring tool to reduce the risk of overtraining, (b) for monitoring athletes’ mood responses to injury, (c) as an index of well-being, and (d) as a catalyst for discussion between athlete and sport psychologist. Future research in a Greek-language context might investigate mood responses to various exercise modalities, assess the effectiveness of mood regulation strategies, link moods to sports performance, or investigate mood profiles as indicators of mental health status. Given its brevity, the BRUMS-Greek may be used to assess mood unobtrusively during pre-competition or during breaks in competition in sporting events that involve several rounds being played over the course of a day, such as rugby sevens or sport shooting.

Regarding the I-PANAS-SF [72] that was translated and used in the present study for concurrent validity purposes, the English-language version of the scale has previously shown cross-sample stability, internal reliability, temporal stability, and cross-cultural factorial invariance among native and non-native English-speaking participants [72]. In the present study, the I-PANAS-SF-Greek data showed a good fit to the existing measurement model, adequate internal consistency was demonstrated, and the hypothesised pattern of correlations with the BRUMS-Greek subscale scores was supported. However, further examination of the psychometric properties of the I-PANAS-SF-Greek is recommended prior to its use in other research contexts.

Limitations are acknowledged related to the heterogeneity of the sample. The mean age of the participants was 35 years, and 75% were aged 18–45 years, with only 25% in the 46–64 years age range. Therefore, the present findings may be more applicable to the younger adult portion of the sample than the older middle-aged group. Also, because women represented 76% of the sample, the present findings may better represent the female portion of the sample. Additionally, because 79% of the sample were exercise participants rather than inactive adults, the present findings may be more applicable to the exercise participants group. The socio-cultural aspects and level of education attained by the participants were not evaluated. Hence, caution should be exercised in generalizing the present findings to the broader Greek population.

Overall, our findings support the factorial validity, internal consistency reliability, concurrent validity, and known-group validity of the BRUMS-Greek. Future investigations may focus on the antecedents, correlates, and behavioral consequences of mood states among exercise participants, physically inactive adults, and sports participants. However, in measuring affect, mood, or emotion [17], the measures should be chosen by researchers with due care, using the recommended three-tiered approach offered by Ekkekakis (pp. 171–172, [17]). Using this approach, researchers should explain (a) which construct is targeted and why; (b) which conceptual model of the chosen construct has been adopted and why; and (c) why the measure chosen best reflects the operationalisation of the components of the chosen conceptual model.

## Figures and Tables

**Figure 1 sports-11-00234-f001:**
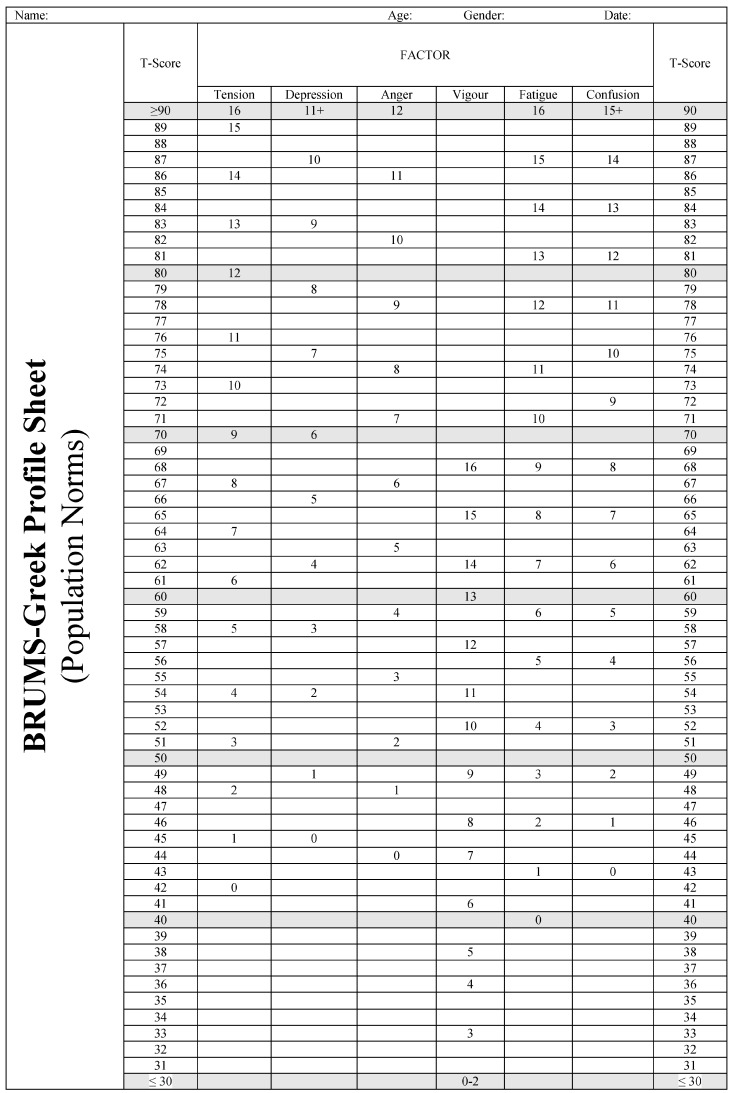
Profile sheet for the Greek translation of the Brunel Mood Scale (BRUMS-Greek), based on 1786 participants.

**Figure 2 sports-11-00234-f002:**
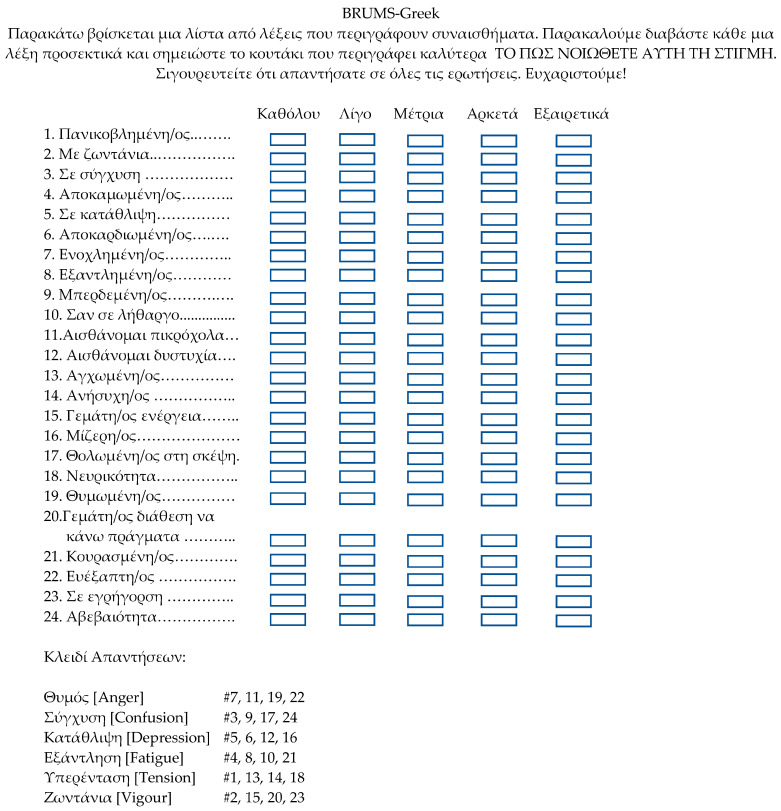
The BRUMS-Greek questionnaire and scoring instructions.

**Table 1 sports-11-00234-t001:** Description of sample characteristics.

Variables		
Sex		
	Males (%)	578 (32.4%)
	Females (%)	1208 (67.6%)
Age (yr)		
	Min.	18
	Max.	64
	*M*	34.73
	*SD*	11.81
	18–25 (%)	511 (28.6%)
	26–35 (%)	543 (30.4%)
	36–45 (%)	316 (17.7%)
	46–55 (%)	310 (17.4%)
	56–64 (%)	106 (5.9%)
Height (m)		
	Min.	1.42
	Max.	2.07
	*M*	1.70
	*SD*	0.09
Weight (kg)		
	Min.	32
	Max.	120
	*M*	70.89
	*SD*	14.19
Body mass index		
	Min.	15.87
	Max.	43.51
	*M*	24.16
	*SD*	3.72
	Underweight (%)	56 (3.1%)
	Normal weight (%)	1099 (61.5%)
	Overweight (%)	499 (27.9%)
	Persons with obesity (%)	132 (7.4%)
Exercise participation		
	Exercise participants (%)	1417 (79.3%)
	Inactive adults (%)	369 (20.7%)

Note. *M*—mean; *SD*—standard deviation; Min.—minimum value; Max.—maximum value.

**Table 2 sports-11-00234-t002:** Descriptive statistics and completely standardised parameter estimates for BRUMS-Greek items (*n* = 1786).

BRUMS Factor	Original Item	Translated Item	*M*	*SD*	Item Skewness	Item Kurtosis	Item Loading	Item Uniqueness	SMC
Anger	Annoyed	Ενοχλημένη/ος	0.44	0.87	2.23	4.65	0.839	0.545	70%
	Bitter	Aισθάνομαι πικρόχολα	0.20	0.62	3.82	16.17	0.843	0.537	71%
	Angry	Θυμωμένη/ος	0.35	0.82	2.71	7.22	0.856	0.517	73%
	Bad tempered	Ευέξαπτη/ος	0.62	1.00	1.63	1.88	0.715	0.699	51%
Confusion	Confused	Σε σύγχυση	0.45	0.86	2.08	3.82	0.803	0.596	64%
	Mixed up	Μπερδεμένη/ος	0.54	0.94	1.88	3.00	0.844	0.537	71%
	Muddled	Θολωμένη/ος στη σκέψη	0.50	0.93	1.98	3.35	0.798	0.603	63%
	Uncertain	Aβεβαιότητα	0.71	1.13	1.55	1.38	0.847	0.532	71%
Depression	Depressed	Σε κατάθλιψη	0.32	0.71	2.54	6.60	0.865	0.502	74%
	Downhearted	Aποκαρδιωμένη/ος	0.32	0.75	2.75	7.77	0.864	0.504	74%
	Unhappy	Aισθάνομαι δυστυχία	0.25	0.69	3.36	12.15	0.899	0.438	80%
	Miserable	Μίζερη/ος	0.24	0.67	3.27	11.66	0.871	0.491	75%
Fatigue	Worn out	Aποκαμωμένη/ος	0.53	0.90	1.82	2.76	0.733	0.680	53%
	Exhausted	Εξαντλημένη/ος	0.98	1.12	0.97	−0.01	0.870	0.492	75%
	Sleepy	Σαν σε λήθαργο	0.39	0.81	2.39	5.69	0.721	0.692	52%
	Tired	Κουρασμένη/ος	1.29	1.17	0.61	−0.58	0.785	0.619	61%
Tension	Panicky	Πανικοβλημένη/ος	0.27	0.73	3.04	9.30	0.663	0.748	44%
	Anxious	Aγχωμένη/ος	0.87	1.12	1.18	0.44	0.887	0.462	78%
	Worried	Aνήσυχη/ος	0.80	1.08	1.29	0.82	0.919	0.394	84%
	Nervous	Νευρικότητα	0.66	1.00	1.58	1.81	0.775	0.633	60%
Vigour	Lively	Με ζωντάνια	2.50	1.03	−0.53	−0.16	0.812	0.584	65%
	Energetic	Γεμάτη/ος ενέργεια	2.36	1.12	−0.41	−0.54	0.911	0.412	83%
	Active	Γεμάτη/ος διάθεση να κάνω πράγματα	2.44	1.14	−0.46	−0.55	0.861	0.508	74%
	Alert	Σε εγρήγορση	2.09	1.19	−0.22	−0.83	0.675	0.738	45%

Note. All factor loadings and item uniquenesses are significant (*p* < 0.05); SMC—Squared multiple correlation.

**Table 3 sports-11-00234-t003:** Descriptive statistics, reliabilities, and inter-correlations among BRUMS-Greek subscales (*n* = 1786).

Subscale	*M*	*SD*	Skewness	Kurtosis	Range	A	1	2	3	4	5
1. Anger	1.62	2.60	2.34	6.56	0–16	0.77	–				
2. Confusion	2.21	3.15	1.79	3.15	0–16	0.82	0.74 *	–			
3. Depression	1.14	2.37	2.99	10.46	0–16	0.85	0.74 *	0.73 *	–		
4. Fatigue	3.21	3.18	1.21	1.25	0–16	0.78	0.54 *	0.57 *	0.55 *	–	
5. Tension	2.62	3.16	1.43	1.71	0–16	0.80	0.69 *	0.79 *	0.65 *	0.53 *	–
6. Vigour	9.40	3.76	−0.36	−0.42	0–16	0.85	−0.23 *	−0.29 *	−0.31 *	−0.34 *	−0.22 *

Note. * *p* < 0.001. α—Cronbach’s alpha coefficient. Subscale scores are created by summing the four items with a possible score range of 0–16.

**Table 4 sports-11-00234-t004:** Descriptive statistics and reliabilities for I-PANAS-SF-Greek subscales, and two-tailed correlations with BRUMS-Greek subscales (*n* = 369).

	I-PANAS-SF Positive Affect	I-PANAS-SF Negative Affect
*M*	3.54	2.44
*SD*	0.66	0.63
Range	1.00–5.00	1.00–4.60
α	0.76	0.70
Anger	−0.26 **	0.46 **
Confusion	−0.35 **	0.50 **
Depression	−0.29 **	0.46 **
Fatigue	−0.25 **	0.39 **
Tension	−0.25 **	0.57 **
Vigour	0.67 **	−0.24 **

Note. ** *p* < 0.01.

**Table 5 sports-11-00234-t005:** MANOVAs of BRUMS-Greek subscale scores by exercise participation, sex, age, and BMI.

Pre- and Post-Exercise (*n* = 398)							
	Pre-Exercise		Post-Exercise				
Subscale	*M*	*SD*	*M*	*SD*	*F*	*η* ^2^ * _p_ *	*d*
Anger	0.86	1.72	0.44	1.44	39.32 ^†^	0.036	0.47 (s)
Confusion	1.20	2.17	0.53	1.51	66.93 ^†^	14.5	0.41 (s)
Depression	0.47	1.40	0.22	1.06	21.55 ^†^	0.052	0.23 (s)
Fatigue	2.36	2.66	2.43	2.62	0.36	0.001	0.02 (vs)
Tension	1.64	2.28	0.66	1.49	108.97 ^†^	0.216	0.52 (m)
Vigour	10.31	3.49	11.06	3.31	20.78 ^†^	0.050	0.22 (s)
Exercise participation (*N* = 1786)							
	Exercise participants (*n* = 1417)		Physically inactive adults (*n* = 369)				
Subscale	*M*	*SD*	*M*	*SD*	*F*	*η* ^2^ * _p_ *	*g*
Anger	1.37	2.38	2.58	3.14	66.04 ^†^	0.036	0.47 (s)
Confusion	1.90	2.95	3.40	3.57	68.62 ^†^	0.037	0.48 (s)
Depression	0.87	2.02	2.14	3.21	87.77 ^†^	0.047	0.54 (m)
Fatigue	2.86	2.99	4.55	3.53	86.13 ^†^	0.046	0.54 (m)
Tension	2.24	2.89	4.07	3.72	103.39 ^†^	0.055	0.59 (m)
Vigour	9.52	3.78	8.97	3.65	6.32	0.004	0.14 (vs)
Sex (*N* = 1786)							
	Male (*n* = 578)		Female (*n* = 1208)				
Subscale	*M*	*SD*	*M*	*SD*	*F*	*η^2^_p_*	*g*
Anger	1.71	2.71	1.57	2.55	1.13	0.001	0.05 (vs)
Confusion	2.12	3.16	2.25	3.14	0.74	0.000	0.04 (vs)
Depression	1.13	2.52	1.14	2.30	0.00	0.000	0.00 (vs)
Fatigue	2.97	3.13	3.32	3.21	4.89	0.003	0.11 (vs)
Tension	2.32	3.04	2.76	3.21	7.67 *	0.004	0.14 (vs)
Vigour	9.80	3.66	9.21	3.79	9.52 *	0.005	0.15 (vs)
Age group (*N* = 1786)							
	≤35 years (*n* = 1054)		≥36 years (*n* = 732)				
Subscale	*M*	*SD*	*M*	*SD*	*F*	*η* ^2^ * _p_ *	*g*
Anger	1.69	2.61	1.52	2.59	1.87	0.001	0.06 (vs)
Confusion	2.49	3.31	2.86	0.71	19.78 ^†^	0.011	0.21 (s)
Depression	1.19	2.44	1.06	2.27	1.44	0.001	0.05 (vs)
Fatigue	3.30	3.20	3.07	3.16	2.22	0.001	0.07 (vs)
Tension	2.80	3.21	2.35	3.07	8.91 *	0.005	0.14 (vs)
Vigour	9.24	3.86	9.64	3.60	4.85	0.003	0.10 (vs)
BMI excluding overweight (*N* = 1287)							
	Underweight/normal weight (*n* = 1155)		Persons with obesity (n = 132)				
Subscale	*M*	*SD*	*M*	*SD*	*F*	*η* ^2^ * _p_ *	*g*
Anger	1.55	2.54	2.15	2.92	6.40	0.005	0.23 (s)
Confusion	2.25	3.13	2.44	3.21	0.44	0.000	0.06 (vs)
Depression	1.08	2.25	1.60	2.69	6.10	0.005	0.22 (s)
Fatigue	3.13	3.13	3.96	3.45	8.13 *	0.006	0.26 (s)
Tension	2.59	3.09	3.15	3.48	3.70	0.003	0.17 (vs)
Vigour	9.42	3.75	9.05	3.76	1.17	0.001	0.10 (vs)

Note. * *p* < 0.008; ^†^ *p* < 0.001. *η*^2^*_p_*—Partial eta squared. Subscale scores are the sum of four item scores ranging from 0–16. vs—very small effect size (<0.20); s—small effect size (0.20–0.50); m—medium effect size (0.50–0.80).

**Table 6 sports-11-00234-t006:** Standardised T-scores for the BRUMS-Greek (*N* = 1786).

Raw Score	Anger	Confusion	Depression	Fatigue	Tension	Vigour
0	44	43	45	40	42	25
1	48	46	49	43	45	28
2	51	49	54	46	48	30
3	55	52	58	49	51	33
4	59	56	62	52	54	36
5	63	59	66	56	58	38
6	67	62	70	59	61	41
7	71	65	75	62	64	44
8	74	68	79	65	67	46
9	78	72	83	68	70	49
10	82	75	87	71	73	52
11	86	78	91	74	76	54
12	90	81	96	78	80	57
13	94	84	100	81	83	60
14	97	87	104	84	86	62
15	101	91	108	87	89	65
16	105	94	113	90	92	68

## Data Availability

The data are available from the corresponding author upon reasonable request.

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
