# Peer review of "A Greek Translation of the Brunel Mood Scale: Initial Validation among Exercise Participants and Inactive Adults"

_sports, 2023, doi:10.3390/sports11120234_

Round 1

Reviewer 1 Report

Comments and Suggestions for Authors

Dear Authors,

Congratulation for a nice article, we particularly appreciate the implications of your findings part where you give very goods perspectives. Sometime you mixture the methodology and the introduction, but reading the whole article, future readers will understand the procedure.

In the lines 79 to 99 of the introduction we suggest differentiate studies related with POMS from the ones related to BURMS. Please finish the introduction, in one phrase, resuming the main objectives and innovative aspects of the study (Validation of BRUMS, potential measure of this questionnaire on positive and negative trait, and mood measure evolution with exercise).

Line 265 we suggest of PANAS trait …

In the methodology we think necessary to clarify in witch basis you consided the participants physically active or not?

Line 454, Please discuss witch kind of exercise your participants did, and the possibility of the global intensity of the exercises being of low intensity and not moderate or Hight (If the activity was pilates the fatigue level should be different that if it was body pump). And what should be the impact of this possibility on the results of your study where the evolution of the fatigue was not significative! In our opinion it could be a limit for this part of the study. May be in the subject that exercised you can see the impact of the fatigue on the results of the questionnaires, to clarify this situation?

Author Response

Responses to the reviewers’ comments

Dear Reviewers

Many thanks for your thoughtful comments on the manuscript with the aim of improving clarity of the text. Responses to comments are provided below.

Reviewer 1

  1. For lines 84 to 102, material was added before each reference number to indicate whether the POMS or the BRUMS had been used in the studies.

  1. A section was added to the end of the Introduction summarizing the objectives of the study (L 159-162).

“Therefore, the main objectives of the study were to provide initial validity evidence for the BRUMS-Greek, to examine links between mood scores and the constructs of trait positive and negative affect, and to advance the use of the BRUMS instrument in relation to exercise behavior.”

  1. Change implemented in line 281.

  1. “Individuals were defined as exercise participants if they exercised in a fitness center. Individuals were defined as inactive if they did not exercise at all for at least a period of 12 consecutive months.” (L. 170-173).

  1. In relation to the comment on fatigue, now L. 521-524. A section was added:

“The exercise participants of this subsample attended indoor group-based exercise programs combining aerobic and resistance exercises, mainly of a moderate to high intensity, and therefore it is reasonable to expect high fatigue scores post-exercise [26]”.

Reviewer 2

Comment 1. A Table with demographic details has been added in the Participants section (Table 1) (L. 175) along a sentence: “Demographic information in terms of sex, age, height, weight, body mass index, and exercise participation is provided in Table 1.” (L. 167-168).

Comment 2. A paragraph has been added to the Discussion section in relation to the suggested limitations of the study:

L.633-642. “Limitations are acknowledged related to the heterogeneity of the sample. The mean age of the participants was 35 years, and 75% were aged 18-45 yrs., with only 25% in the 46-64 yrs. age range. Therefore, the present findings may be more applicable to the younger adult portion of the sample than the older middle-aged group. Also, because women represented 76% of the sample, the present findings may better represent the female portion of the sample. Additionally, because 79% of the sample were exercise participants rather than inactive adults, the present findings may be more applicable to the exercise participants group.  Socio-cultural aspects and level of education attained by the participants was not evaluated. Hence, caution should be exercised in generalizing the present findings to the broader Greek population.”

Reviewer 2 Report

Comments and Suggestions for Authors

General Comments

This is an interesting study, which aimed to provide initial validity evidence of a Greek translation of the 24-item Brunel Mood Scale (referred to as the BRUMS-Greek) for evaluation of anger, confusion, depression, fatigue, tension, and vigour. The Authors  hypothesised that the 24-item translated-into-Greek BRUMS scores would support a six correlated factor measurement models in line with previous translated versions of the scale, and that the BRUMS-Greek factors would display adequate internal consistency reliability. A total 1,786 Greek adults (68% women) aged 18–64 years (77% young and 23% middle-aged) participated in this study. From them 79% were exercise participants and 21% physically inactive. The results of this study supported the hypothesis of six correlated-factor measurement model, whereas internal consistency reliability of BRUMS-Greek subscales was supported via Cronbach alpha coefficients. More positive and less negative moods were reported immediately after participating in an acute indoor group exercise compared to pre-exercise mood, whereas exercise participants reported more positive and less negative mood states compared to physically inactive subjects. Women reported higher tension and lower vigour scores than men. Tension scores were higher and confusion scores lower among younger participants (≤ 35 years) and obese subjects reported higher negative mood scores as compared to normal weight subjects. The Authors concluded that the BRUMS-Greek demonstrated acceptable psychometric characteristics, and is proposed to be a suitable measure for use with exercise participants, physically inactive adults, and other Greek populations to explore research questions related to mood.

The manuscript is generally well written. However, the design of this paper should be little improved.

1. I suggest to add a table with demographic data (incl. gender, age groups, BMI, sport participation) on Materials and Methods, Participants (Page 4) for better presentation of the measured participants.

2. The authors should be to include limitations of this study at the end of the Discussion

(1) The first limitation relates to the heterogeneity of age of the participants. The mean age of the participants was 35 years, whereas 77% were young subjects (aged 18-45 years) and 23% middle-aged (aged 46-64 years). The older middle-aged group (aged 56-64 years) represented only 6% of the sample size. It can be concluded that this study is valid predominantly for young adults.

(2) The second limitation relates to the gender representation of sample, which included 68% of women, ie. This study is valid predominantly in female adults.

(3) The third limitation relates to sport participation – 79% were exercise participants and 21% physically inactive, ie. This study is valid predominantly in physically active adults.  

(4) The fourth limitation relates to socio-cultural aspects and education, which was not evaluated in this study.

Finally, these limitations may restrict the ability to generalize the results of this study to the broader Greek population.

Specific Comments

 2. Materials and Methods

2.1. Participants

Page 4. Please add a table a table with demographic data of the participants (see General Comments).

4. Discussion

Page 16. Please describe limitations of this study at the end of Discussion (see General Comments).

Author Response

(The authors gave the same response as above.)
